# Understanding the Response of Canadians to the COVID-19 Pandemic Using the Kübler-Ross Model: Twitter Data Analysis

**DOI:** 10.3390/ijerph20043197

**Published:** 2023-02-11

**Authors:** Ruth G. Abebe, Schwab Bakombo, Anne T. M. Konkle

**Affiliations:** 1Interdisciplinary School of Health Sciences, University of Ottawa, Ottawa, ON K1N 6N5, Canada; 2School of Psychology, University of Ottawa, Ottawa, ON K1N 6N5, Canada; 3Brain and Mind Research Institute, University of Ottawa, Ottawa, ON K1N 6N5, Canada

**Keywords:** COVID-19, Canada, pandemic, stress, policies

## Abstract

Coronavirus disease 2019 (COVID-19) was declared a pandemic in March 2020, which led to the implementation of non-pharmaceutical interventions that had inadvertent physical, mental and social effects. The purpose of this retrospective study was to examine the experiences and responses of Canadians to these interventions on Twitter using the Kübler-Ross Change Curve (KRCC) during the first six months of the pandemic. Tweets were analyzed using sentiment analysis, thematic content analysis and KRCC. The findings highlight that many Canadians attempted to adapt to the changes but had a predominantly negative outlook on policies due to the financial and social repercussions.

## 1. Introduction

The definition of health is often redefined to be more encompassing of the many components that contribute to one’s well-being. Health is described as an overall state of physical, mental, emotional, economic and social well-being [1]. Mental health is regarded as an essential resource that improves an individual’s self-perception, increases their likelihood of achieving success and allows them to positively contribute to society [1]. When people experience poor mental health, this can significantly impact morbidity and mortality [2]. Social support is a crucial factor in reducing the impact of stress and helps to improve perceived mental health [3]. Less social support is correlated with the incidence of negative coping mechanisms during stressful situations and can exacerbate poor mental health [3].

On March 11, 2020, the World Health Organization (WHO) declared the coronavirus disease 2019 (COVID-19) a global pandemic [4]. To reduce the spread of the virus, governments implemented non-pharmaceutical interventions (NPIs) such as quarantine, school closures and personal protective behaviors which included avoiding gatherings, reducing personal contacts and meetings, and increased hand hygiene [4]. These interventions increased physical isolation, school closures and, inadvertently, widespread job losses [4].

While these measures were put in place to protect people from an unknown virus, this had unintended individual, social, and economic repercussions worldwide [4]. Canada in particular ranked as having some of the most restrictive public health restrictions in the world with long-lasting closures of businesses, remote work and gathering restrictions [5]. While these measures may have helped reduce the number of COVID-19 cases compared to countries that had looser restrictions, such as the United States, these restrictive measures may have resulted in long-lasting economic and social impacts as well as stress due to the difficulty adapting to these measures [5]. The WHO released a report that disclosed an increase in substance misuse and direct neurological consequences such as stress, burnout, depression and post-traumatic stress disorder (PTSD) since the pandemic began [4]. In addition to the cognitive impacts of social distancing, the frequency of news reports regarding the high number of cases and fatalities has likely had an impact on the stress levels of many Canadians [4].

Resilience research states that when people have more control and stability during a stressful event, they are more capable of coping with stress [6]. As mental health and substance use are often correlated with employment and financial stability, the policies that governments put in place regarding the Canadian economy are likely to have a significant impact on the mental health and well-being of Canadians. While the COVID-19 vaccine rollout in Canada began in January 2021, a possible return to normal life was envisioned; however, the long-term stress that people experienced during this pandemic has likely had negative consequences that may have long-lasting effects on their mental health.

The effects of past epidemics and pandemics on mental health have been studied extensively and highlight the lasting negative impacts they can have. During the 2009 H1N1 pandemic, researchers found an increased incidence and prevalence of depression and distress among the general population, family members of patients and healthcare workers [7,8]. In the United States, people reported higher levels of anxiety, uncertainty and engaged in more risky behaviors, such as smoking, drinking, recklessness and unsafe work practices during the H1N1 pandemic [8].

During the 2003 SARS epidemic, it was found that individuals in China and Taiwan who were quarantined and experienced the perceived threat of SARS had a high incidence of depressive symptoms and were at an increased risk of developing mental health disorders. There was also a statistically significant increase in suicide rates in the years following the epidemic [9]. In one cohort, the cumulative post-SARS incidence of mental health disorders was 58.9%, while during the post-illness stage, the point period prevalence of PTSD was 32.2%, depression was 14.9% and anxiety was 14.8% [9].

### 1.1. Kübler-Ross Model

The Kübler-Ross model was designed by physician Dr. Elisabeth Kübler-Ross based on the observation that terminally ill patients and their relatives experienced similar emotions during bereavement [10]. These include denial, anger, bargaining, depression and acceptance [10]. This model has been adapted to reflect stages of personal and organizational change as the Kübler-Ross Change Curve (KRCC) [11]. It can have up to seven stages: shock, denial, frustration, depression, experimentation, decision and integration [12]. This model was selected for this study as each stage of the KRCC indicates not only the sentiments that an individual feels when experiencing change, but also provides a model of the progression towards integration with the change. This can help researchers understand the efficacy of change management strategies and use participant responses to reduce stress and promote better adaptation to the change in the future. This model may help in analyzing how loss and change during the pandemic may affect the self-rated stress and mental health of Canadians.

### 1.2. Social Media

In recent years, social media has become an important tool for self-disclosure, social support and encouragement, particularly with the long-term use of physical distancing measures throughout the COVID-19 pandemic [13,14]. Researchers agree that social media is the medium through which people share their lives. It has become the ultimate “online community”, whereby communication, information, news, views, opinions, perspectives, ideas, knowledge, feedback, and experiences related to pandemics, epidemics, viruses, and disease are discussed by the public at large [15,16,17,18]. Twitter is increasingly used as a source of data for research due to the public availability of information in real-time [13]. As a result, Twitter may serve as a useful tool in understanding the effects of the pandemic and the policies surrounding it.

## 2. Purpose

The purpose of this retrospective study was to analyze the statements that Canadians made on Twitter regarding their experiences with the perceived threat of COVID-19 and the non-pharmaceutical interventions implemented during the pandemic. The tweets were analyzed to determine the presence of prominent themes and sentiments using the stages of the Kübler-Ross Change Curve to assess the stress resilience and adjustment of Canadians during the pandemic, as these are essential components of developing healthy coping mechanisms and positive long-term mental health [3].

## 3. Significance

It is predicted that the content of the tweets may be helpful in assessing whether people reach the critical stage of acceptance or integration in the Kübler-Ross model over several months. The responses may shed light on public perception regarding the appropriateness of government interventions and supports. The sentiment regarding government interventions could highlight if the interventions helped Canadians during the pandemic, how their mental well-being may have been affected and possible recommendations to COVID-19 policies to promote resilience.

## 4. Methodology

### 4.1. Ethics Approval

This study did not require ethics approval as it collected publicly available information from Twitter. To protect the privacy of Twitter users, no identifiable or personal information is presented in this study.

### 4.2. Data Extraction

Using the advanced search feature of Twitter, a search was conducted using keywords from five categories: COVID-19, Canadian policies, finances, non-pharmaceutical interventions and mental health pertaining to COVID-19 (Table 1). Within the advanced search feature, the date was specified for tweets that were posted from March 2020 to August 2020. Tweets were then extracted and analyzed using Microsoft Excel. Inclusion criteria for the tweets were those that were tweeted from Canada, included at least one hashtag from the COVID-19 category and one other category, tweets in English, and tweets dated from March 2020 to August 2020. In addition, to reduce the likelihood of including tweets from Twitter bots in the study, tweets and user information were further reviewed and excluded if the contents of the tweet or user profile appeared to belong to a Twitter bot.

Tweets were manually geolocated by reviewing tweets and Twitter user profiles for location information to ensure that the included tweets were from users living in Canada and impacted by Canadian policies regarding COVID-19 during the first six months of the pandemic.

### 4.3. Sampling Process

Figure 1 shows the breakdown of the tweet selection, the population and sampling process. From the inclusion criteria, a population of 593 tweets was generated. Those tweets were further reviewed to remove non-relevant tweets that did not have enough information to pull sufficient sentiment or thematic content from as well as those that appeared to belong to Twitter bots. Duplicate tweets were also removed from the population to give a remaining number of tweets at 459. A random sample of 1/3 of the 459 remaining tweets was drawn which created a sample of 153.

### 4.4. Data Analysis

From this sample, a sentiment analysis was conducted using the Microsoft Excel Azure Machine Learning add-in. These tweets received a sentiment ranging from positive, negative, neutral or mixed. These sentiments were then manually verified to ensure accuracy of the analysis.

A thematic content analysis was conducted using NVivo 11. The sample population was coded by month, and using a bag-of-words model, a query was performed to tokenize tweets and determine common words. These words were grouped together by similarity to determine prominent themes in the tweets. Once the themes were extracted, they were compared with the sentiments that were expressed in each month to determine the strongest sentiments associated with each theme.

In addition, the Kübler-Ross Change Curve was used to assess which stage of the KRCC the user was experiencing at the time of the tweet. Using the existing bag-of-words model, each tweet was reviewed and assigned to the appropriate stage(s) of the curve based on the words that best fit each stage of the KRCC. Combining sentiment analysis, thematic content analysis and the KRCC, tweets were analyzed to determine how Twitter users were experiencing changes associated with the pandemic based on each stage of the KRCC, the themes that were strongly associated with their reactions and whether they had positive, negative or neutral sentiments towards those themes. Based on when the tweets were posted, the KRCC demonstrates the progression towards interacting with change and whether there are indications of stress and potential negative impacts.

The accounts were also assessed to determine whether the user was an individual, organization, researcher or news reporter. The sentiments, themes and KRCC stage were compared by month to determine whether there were trends between March and August 2020. The content of the tweets was analyzed to identify possible impacts of COVID-19 response policies on mental health and to identify opinions regarding the efficacy and repercussions of COVID-19 response policies and how this impacted the resilience and stress levels for Canadians.

## 5. Results

This study explored issues that were most prevalent among Canadian Twitter users and how they navigated them during the first six months of the pandemic. The sentiment analysis and thematic content analysis identified the theme(s) within the tweets and the sentiment that was associated with them. The Kübler-Ross Change Curve identified what stage of the curve the users were likely experiencing when they wrote the tweet. Researchers agree that positive and negative sentiments interact [19]. A sentiment analysis on YouTube showed that many negative comments originate from users who are more inclined to post hateful comments without constructive criticism [19]. Our findings suggest that it is often the case with tweets. Negative comments are often completely disconnected from the actual content of a video, without offering any constructive criticism to the author, so that the latter may improve their technique [19]. In the context of our analysis, it is evident that even though individuals bear the brunt for most negative tweets, organizations, the media, and researchers are also among those unleashing such sentiments.

The sentiment analysis found that 71/153 (46.4%) of tweets expressed a negative sentiment. Among the negative sentiment tweets, 51/71 (71.8%) belonged to individuals, 12/71 (16.9%) belonged to organizations (for-profit and non-profit), 5/71 (7.0%) belonged to news reports and 3/71 (4.2%) belonged to researchers. This shows that most of the tweets came from individual accounts and were a good representation of the general Canadian public.

The four categories that were referenced the most from March to August were Canadian policies, financial concerns, impacts of NPIs, and direct health impacts. Table 2 shows the percentages of tweets with a negative sentiment that had these categories.

### 5.1. Analysis Using the Kübler-Ross Change Curve

The Kübler-Ross Change Curve was a critical tool in understanding the stages of adaptation that Canadians were experiencing throughout the many changes of the pandemic. There are seven categories: shock, denial, frustration, depression, experimentation, decision and integration. The tweets analyzed did not display integration as data collection occurred early during the pandemic and integration may take longer than the six-month timeline of this project. The stages that were present in the sample are shown in Figure 2A–F.

### 5.2. Shock

Shock is the initial stage of the KRCC when a person is faced with an unexpected change [11]. From the sample tweets in March, 5/28 (17.6%) of the tweets expressed shock (Figure 2A). After March, the number of tweets that expressed shock continued to decrease in the following months and by August only 2/29 (6.9%) expressed shock (Figure 2A).

### 5.3. Denial

Denial is the second stage of the Kübler-Ross model when a sudden change can become difficult to believe or accept and this is expressed through disbelief by the individual [11,12]. In March 1/28 tweets expressed denial (Figure 2B). While in the following months, the number of tweets that expressed denial increased. In April there were 4/29 (13.8%) (Figure 2B). In the following months, the percentage of tweets with denial decreased to 6.9% in August (Figure 2B).

### 5.4. Frustration

Frustration is the third stage of the Kübler-Ross model when an individual recognizes the change and can become upset or angry because of it [11,12]. In March 4/28 (14.3%) expressed frustration (Figure 2C). These tweets expressed frustration with the new challenges that resulted because of COVID-19. For example, because of the work from home policies put in place due to COVID-19, many people experienced slow broadband internet service, especially those living in rural areas. By May, the number of tweets expressing frustration grew to 10/33 (30.3%) (Figure 2C). These tweets mentioned issues that had worsened since the pandemic and the implementation of NPIs such as the increase of domestic abuse, business bankruptcies and unemployment, and high COVID-19 mortality rate in long-term care homes. By August 6/29 (24.1%) tweets expressed frustration, specifically surrounding issues with finances (Figure 2C).

### 5.5. Depression

Depression is the fourth stage of the Kübler-Ross model when an individual feels overwhelmed with the new change and lacks the energy to interact with the change [11,12]. In March, there were few tweets that expressed depression (6.9%). These were mainly concerned with the deaths that resulted as a result of COVID-19 (Figure 2D). While the number of tweets that expressed depression decreased in April, this number increased in the months that followed. In May 5/33 (15.2%) tweets expressed feelings of depression (Figure 2D). Some individuals also expressed sadness surrounding the number of people who had died due to COVID-19 and depression over financial concerns and unemployment. One person wrote, “People are losing their jobs, freedom, and ability to socialize and see loved ones”. Tweets that expressed depression continued to increase in June and July (Figure 2D). In August, there were 7/29 (17.9%) that expressed depression (Figure 2D). These tweets expressed more concern about the impending second wave and feeling overwhelmed due to financial difficulties.

### 5.6. Experimentation

Experimentation is the fifth stage of the Kübler-Ross model when an individual begins engaging with the new situation [11,12]. In March, experimentation was the most common KRCC stage in tweets with 13/28 (46.4%) expressing it and 5/13 having a positive sentiment (Figure 2E). The number of tweets with experimentation decreased in the following months with the tweets in July containing the fewest reference to experiment (Figure 2E). In August, there were 9/29 (31.0%) tweets that expressed experimentation (Figure 2E). These focused on encouraging the Canadian government to expand COVID-19 response policies to support more Canadians and propose mandatory COVID-19 testing.

### 5.7. Decision

Decision is the sixth stage of the Kübler-Ross model when an individual is learning to interact in the new situation and is feeling more positive about the future [11,12]. In March 3/28 (10.7%) tweets demonstrated decision (Figure 2F). These tweets decreased after March, with the fewest in June (6.5%) (Figure 2F). In August 4/29 (13.8%) tweets demonstrated decision (Figure 2F). One tweet stated, “Shoutout to the thousands of #publicservants across #Canada that continue to work to #help #Canadians in a variety of ways”.

### 5.8. Integration

Integration is the final stage of the Kübler-Ross model when an individual has fully accepted and integrated with the change and becomes a renewed person [11,12]. Throughout this study, no tweets expressed integration from March to August 2020 as this was still too early in the pandemic to potentially see this stage of the KRCC.

## 6. Discussion

### 6.1. Themes

The themes that were present in the tweets were based on six categories: Canadian policies, financial concerns, impacts of NPIs, direct health impacts of COVID-19, research, and support and encouragement. Tweets categorized as Canadian policies were ones that referenced policies surrounding the COVID-19 response, mentioned the actions of politicians or compared policies in Canada to those of other countries. Financial concerns were tweets that mentioned the effects of economic policies, such as the Canadian Emergency Relief Benefit (CERB) and Employment Insurance (EI) and stress due to financial difficulties experienced directly and indirectly from the pandemic. Impacts of non-pharmaceutical interventions (NPIs) referred to the effects of physical distancing measures in place such as school closures, isolation/quarantine, and the importance of personal protective behaviors. Direct health impacts were related to concerns regarding the health and safety of citizens, the disclosure of those who experienced COVID-19-related symptoms and impacts of physical or mental health issues that may have resulted indirectly from the pandemic. Research referred to tweets that shared new information on research advancements in understanding COVID-19 and its impacts, vaccine development, and treatment. Support and Encouragement referred to tweets that shared words of encouragement for others and provided support for people.

### 6.2. Opinions on Canadian Policies

This theme received the most negative remarks on Twitter from March to August 2020. Most users expressed that government policies were not implemented early enough to protect people from COVID-19. Some referenced that while Canada was not prepared for the 2003 SARS epidemic, the government should have been better prepared for the COVID-19 pandemic 17 years later.

Canada made sweeping reforms to infectious disease control after SARS, such as the creation of the Public Health Agency of Canada, a stronger federal presence in public health, enhanced surveillance, and increased preparation and equipment for infection control [20,21]. While Canada introduced a lockdown soon after the pandemic announcement by the WHO, the policies in place did not protect vulnerable groups, particularly those in long-term care homes [20,21]. One Twitter user described frustration that more thorough inspection of long-term care facilities should have been conducted to identify issues with infection prevention and control. Those brought in to inspect these facilities highlighted that basic health care and infection prevention were not being conducted in many facilities [20]. In addition, it was found that the mortality rate in long-term care facilities in Canada was higher than 16 other countries belonging to the Organization for Economic and Co-operation and Development (OECD) [20].

### 6.3. Financial Concerns

The major subthemes in this category were opinions on the efficacy of the Canadian Emergency Response Benefit, experiences with unemployment and the changing work landscape, and stress from financial difficulties. To reduce the spread of the virus, many retail stores and small businesses were required to close during lockdowns. This led to many lay-offs and small business owners being forced to close, while big businesses such as Walmart, Costco and Amazon were allowed to continue operating and saw more profit [22]. While CERB and the Canadian Emergency Business Account (CEBA) were available to business owners, they expressed that it was not enough money for them to support themselves and their families. One tweet stated, “#CEBA is still leaving #soleproprietors in the cold.”

Some Twitter users expressed that CERB would discourage people from seeking permanent employment and allowed them to take advantage of taxpayer dollars. Some tweets stated that lower income groups would use this as an opportunity to not seek employment. One person tweeted, “Why work? When #refugees arrive in #Canada, they hit the jackpot. No need to work”. Contrary to this belief, the economic impacts of the pandemic disproportionately affected refugees and immigrants [22]. Statistics Canada found that members of these groups were much more likely to lose their jobs than Canadian-born workers because they often work low-paying jobs with less opportunity to work from home and at increased risk of being replaced by automation [22].

While CERB and other governmental financial aid temporarily helped people who experienced job loss, this brought up the debate of universal basic income (UBI) in Canada. UBI is an income paid to individuals regardless of employment status or education [23]. As the financial concerns brought on by the pandemic were difficult for many people of lower income groups, many Twitter users felt that CERB could be expanded upon to become UBI to improve the social safety net in Canada and protect vulnerable groups from poverty.

### 6.4. Impacts of NPIs

Within this theme, the major subthemes that were present were disclosure regarding isolation, the efficacy of NPIs and coping with NPIs. In March, while many tweets were encouraging people to follow public health protocols, by May when the number of COVID-19-positive people continued to increase, people felt that Canada had not learned from SARS and could have kept the number of community cases lower if swift action had been taken to reduce transmission. In August, all tweets about NPIs were negative. Some of these tweets expressed a belief that the government did not do enough to curb the spread of COVID-19, that NPIs were unfair and ineffective, and that social isolation was having negative effects on the mental health and well-being of Canadians.

A cross-sectional study on the mental health of people that experienced isolation and quarantine during the beginning of the COVID-19 pandemic in China found that those who were isolated experienced high rates of depression, anxiety, and stress, especially during the first week of isolation [24]. Further research found that longer isolation was associated with worse mental health outcomes due to prolonged stress as the mental health of people that experienced isolation and fear of the virus was more negatively impacted than the control group, even several years after the epidemic or pandemic had ended [7,24,25].

Twitter users also highlighted that due to the confinement required by lockdowns, many cases of domestic abuse increased during the pandemic. This left many victims of abuse unable to leave their abuser and seek help [26]. For example, one tweet stated, “Many victims have lost access to telephones or computers, including children who may be in danger”. In addition, due to NPIs, there was a greater demand for telework and online learning, which placed a high strain on broadband internet. Many people living in rural communities often faced issues with high-speed internet and this made it especially difficult for people to work from home. This further highlights the inequities that exist with the access to technology in Canada [27].

### 6.5. Direct Health Impacts

This theme was present in tweets that described the number of active cases of COVID-19, the death toll, people’s personal experiences with COVID-19, and possible long-term health effects. In March, the majority of tweets were concerned with flattening the curve and reminding people to follow public health protocols to reduce the number of active cases. One tweet listed the number of people that died from COVID-19 in March from various countries around the world. The tweets from March to August showed a shift in the countries that had the highest active cases and death rates. In the very beginning of the pandemic China and Italy were leading the world in deaths from COVID-19, but this quickly changed with the United Kingdom, the United States and Canada seeing a rapid surge in COVID-19 transmission.

By May 2020, when lockdown restrictions started to ease, some tweets expressed concern about a second wave because at the time it was not known where the source of the transmission was coming from, whether it was community or travel, and because COVID-19 testing was not mandatory. One tweet stated, “B.C. could see 2nd wave of COVID-19 in September bigger than the first. Do not forget to wear your masks and sanitize regularly.”

Another concern with COVID-19 as expressed by 4/33 of the direct health impact tweets (Table 2), was the concern about how COVID-19 may affect other underlying conditions such as gestational diabetes and stroke. By August 2020, 3/6 tweets had a negative sentiment and expressed concern about people not following COVID-19 protocols, not getting tested and whether people would be vaccinated once the vaccine was made publicly available (Table 2).

As SARS-CoV-2 is a relatively new virus, there is still much being researched about its long-term effects. Since COVID-19 manifests differently in different people and is still not well understood, it is still feared by many Canadians. The fear of the virus coupled with isolation during the pandemic has likely had a significant impact on the mental health and well-being of many Canadians.

### 6.6. KRCC—Shock and Denial

Shock and denial were the most common stages that were present in the first couple of months of the pandemic. This aligns with the KRCC when the change is first introduced and participants are interacting with it [11,12]. These tweets expressed shock at the NPIs implemented and Canada’s Economic Response Plan. In March, there was considerable shock concerning the NPIs as people were unaccustomed to the sudden isolation. There was also shock at the reactions of others after the announcement of the pandemic and NPIs. This included stocking up on essential items for fear that they would run out in stores.

Denial is the next stage and usually comes when the participant has had some time to analyze the change and develop opinions about it [28]. The contents of those tweets were expressing distrust in government policies with the belief that NPIs were unfair and ineffective. In April most of the tweets that expressed denial continued to believe that the virus was not dangerous and mistrusted the government for implementing NPIs.

### 6.7. KRCC—Frustration and Depression

Frustration is the third stage in the KRCC when people are attempting to interact with the new system and trying to reduce the stress response [10,28]. While it is understood that frustration is a normal part of the learning and adjustment period, studies show the importance of a strong support system during this time, especially with the unprecedented length of time that restrictive measures were in place [12,28]. To manage the frustration period, it is critical for people that are experiencing this stage to feel they are supported by government, employers, educational institutions and other organizations. While many efforts were made to help people during the pandemic, it is important that there is equal access to a wide variety of resources to help people that may be feeling frustrated, as this can have a significant impact on preserving mental health and resilience.

Depression is the fourth stage when participants may lose hope in the change, especially if they are not seeing its efficacy [28]. The classroom study using the KRCC conducted by Malone et al. [28] found that to manage depression, it was important for individuals to provide their feedback about the new system and to incorporate their suggestions where possible. Providing a space for Canadians to provide their feedback during this pandemic may have proven to be helpful as some policies may not have been as effective as desired. Therefore, if government officials and policymakers conducted more comprehensive policy impact assessments by incorporating the opinions of the Canadian public, this would allow for more specific and targeted support to be offered and may help people to better adjust to the changes.

### 6.8. KRCC—Experiment, Decision and Integration

Experiment is the fifth stage that people may experience in change management as they start to see the benefits of the change and stress levels decrease [10]. These tweets demonstrated that early in the pandemic, many people were attempting to interact with the changes brought on by the pandemic. This included encouraging people to comply with the public health protocols, informing people about the new services and supports that the government was providing, and writing encouraging messages to people while they were at home. Studies show that experimentation is an important stage that should be rewarded to encourage people to continue working towards integration [12,28].

The analysis demonstrates that the sixth stage, decision, was initially higher in April and increased again in August. Tweets that demonstrated decision included how people adjusted to the changes, the scientific accomplishments in treating and preventing COVID-19 and optimism about the future. These expressed gratitude to public servants and frontline workers that helped people throughout the pandemic and appreciation towards businesses that adjusted to the pandemic, such as becoming online retailers.

While this study was conducted at the beginning of the pandemic and was not able to see people’s progression towards integration, if continued support is provided to people throughout the adjustment period, it can be expected that people will be able to reach integration and have a positive outlook on the experience. It is important to note that reaching integration is contingent upon the level of support that people receive and resilience they can develop during the adjustment period and may mean increasing the support available [12,28].

### 6.9. Limitations

This study looked at the tweets from the first six months of the pandemic. Therefore, one limitation of the dataset is that it does not capture any issues that have emerged since August 2020; especially related to vaccination which we thought would confound the current results. As the pandemic has been rapidly changing, follow-up studies are needed for tweets that may be related to various changes in policies from lockdowns to vaccination mandates after August 2020. Another limitation is that the data were gathered exclusively from Twitter and may not be representative of the entire population of Canada. It is also important to note that Twitter has accounts that are operated by Twitter bots and while measures were taken to reduce the likelihood of including these tweets in the sample, it is difficult to be certain that we were 100% effective. In addition, in order to ensure that the included tweets belonged to Twitter users in Canada, a manual geolocation process was undertaken using public profile information; however, this information is not always reliable or available.

## 7. Conclusions

This study analyzed Twitter data to understand the perspectives of Canadians to the COVID-19 response in Canada. While the government attempted to implement swift action to reduce the spread of COVID-19 early during the pandemic, many Twitter users expressed that it was not effective in containing the virus. Based on the KRCC, the level of impact that the pandemic had on Canadians determined how well they were able to adjust to the change. These findings demonstrate that for more Canadians to be able to adjust to changes and maintain good or excellent mental health, there may need to be more supports made available to help people adapt, achieve their highest potential, and positively contribute to society, especially the most vulnerable in communities. These data also speak to the communication available to Canadians by different governmental bodies and how well messages may or may not have been related to an individual’s real life.

## Figures and Tables

**Figure 1 ijerph-20-03197-f001:**
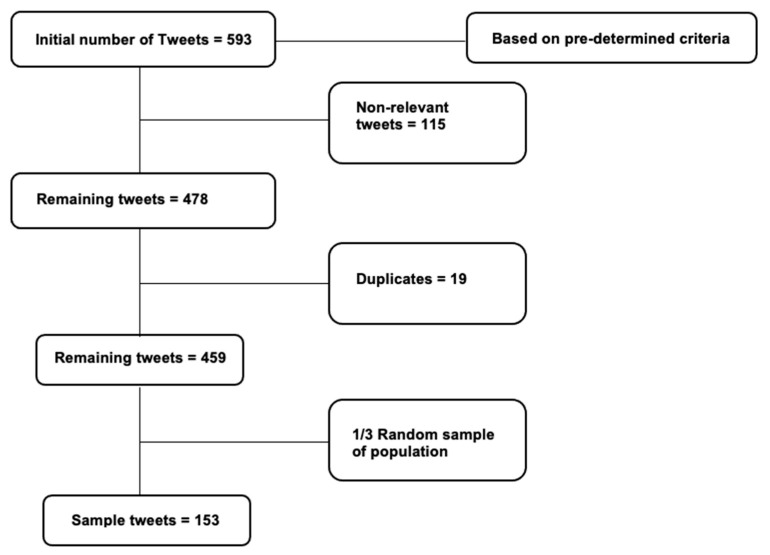
Flow diagram of the selection and sampling process.

**Figure 2 ijerph-20-03197-f002:**
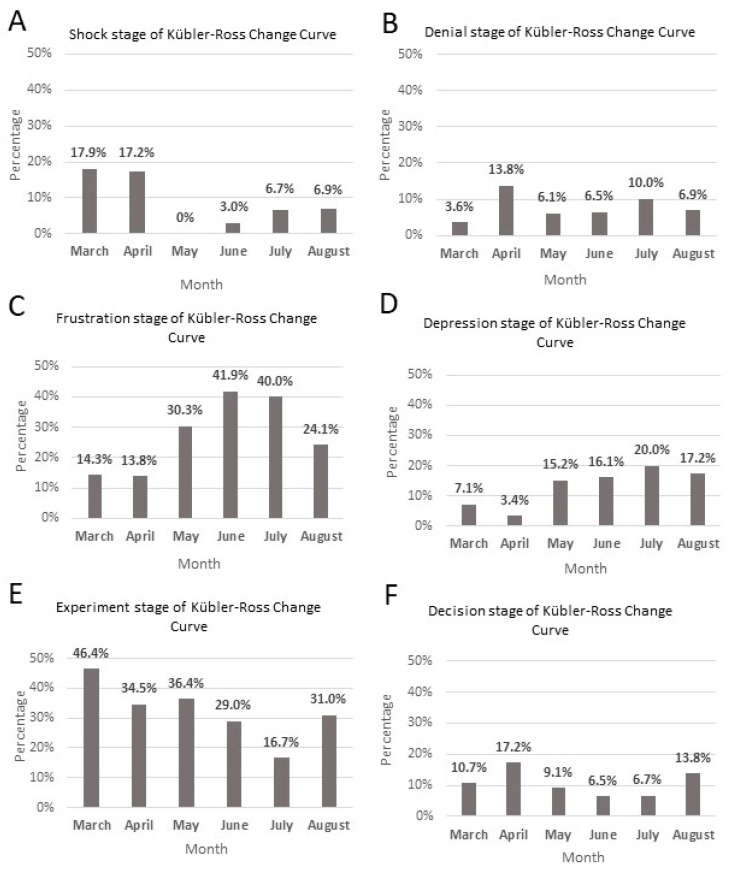
Kübler-Ross Change Curve analysis of tweets that displayed shock (**A**), denial (**B**), frustration (**C**), depression (**D**), experiment (**E**) and decision (**F**) from March to August 2020.

**Table 1 ijerph-20-03197-t001:** Search keywords by category.

COVID-19	Canadian Policies	Finances	Non-Pharmaceutical Interventions	Mental Health
COVID-19 COVID-19 COVID coronavirus SARS-CoV-2 pandemic	Trudeau Ford Canada canada policies capoli COVID-19 Ontario COVID Canada	CERB EI jobs economy unemployed layoffs	lockdown social distancing masks quarantine work from home stay at home	shock stress fear isolation mental health COVID anxiety

**Table 2 ijerph-20-03197-t002:** Breakdown of most common themes with negative sentiment by percentage March–August 2020.

Themes	Percent in March (%)	Percent in April (%)	Percent in May (%)	Percent in June (%)	Percent in July (%)	Percent in August (%)
Canadian policies	25.0	25.0	41.2	13.3	37.5	20.0
Financial concerns	16.7	25.0	5.9	26.7	25.0	45.0
Impacts of NPIs	41.7	41.7	35.3	26.7	18.8	20.0
Direct health impacts	8.3	8.3	17.6	33.3	18.8	15.0

## Data Availability

Publicly available data were analyzed in this study.

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
