# Peer review of "Understanding the Response of Canadians to the COVID-19 Pandemic Using the Kübler-Ross Model: Twitter Data Analysis"

_ijerph, 2023, doi:10.3390/ijerph20043197_

Round 1
Reviewer 1 Report
The paper is titled – “UNDERSTANDING THE RESPONSE OF CANADIANS TO THE COVID-19 PANDEMIC USING THE KÜBLER-ROSS MODEL: TWITTER DATA ANALYSIS”. The purpose of this study was to examine the experiences and responses of Canadians to interventions on Twitter using the Kübler-Ross Change Curve (KRCC) during the first six months of the pandemic. Tweets were analyzed using sentiment analysis, thematic content analysis, and KRCC. The findings highlight that many Canadians attempted to adapt to the changes but had a predominantly negative outlook on policies due to the financial and social repercussions. The work seems novel. However, the presentation of the paper needs improvement. It is suggested that the authors make the necessary changes/updates to their paper as per the following comments:
1. The relevance of using the Kübler-Ross Change Curve (KRCC) isn’t outlined. Why wasn’t a multilabel classifier developed to detect the six basic emotional states such as happiness, sadness, anger, frustration, disgust, and anxiety? How does this KRCC approach work? Does it use a bag of words model to detect various stages: shock, denial, frustration, depression, experimentation, decision, and integration?
2. The purpose of this study was to analyze the statements that Canadians made on Twitter regarding their experiences with the perceived threat of COVID-19 and the non-pharmaceutical interventions implemented during the pandemic. Why does this study focus on Canada and not some of the other countries, such as USA, India, etc., which had a significantly high number of COVID-19 cases as compared to Canada?
3. In Section 1.2, the authors briefly review a couple of works that focused on the analysis of Tweets related to COVID-19. Reviewing just 2 works isn’t sufficient as there have been several works done in this field since the outbreak of COVID-19, such as https://doi.org/10.3390/covid2080076 and https://doi.org/10.1111/josi.12535. Consider citing these recent papers as well as at least 2-3 more papers in this review section.
4. Section 4.2 should be rewritten to clearly discuss the steps for data mining. Specifically, please discuss how these tweets were mined. Was the standard search feature or the advanced search feature of Twitter used, and why?
5. The authors state – “Tweets were manually geolocated”. Please discuss the specifics of this process. Did the authors search for the location information of each Twitter user? How did they take into account that some of these Tweets could not be from real users (i.e., the Tweets may have been written by bots)?
6. The sample size (from Figure 1) seems very less. Consider increasing the number of search words to repeat this study on a greater sample size.
7. In Section 5, the authors discuss the percentage of negative tweets. Classifying tweets into positive and negative sentiment classes isn’t new, and there have been several research works done in this field that focused on a similar classification of Tweets. Please rewrite this section to clearly highlight why this approach should be considered novel.
8. Is the dataset of Tweets available on a dataset storage/search platform such as IEEE Dataport so that the study may be repeated by other researchers in this field?
Author Response
The paper is titled – “UNDERSTANDING THE RESPONSE OF CANADIANS TO THE COVID-19 PANDEMIC USING THE KÜBLER-ROSS MODEL: TWITTER DATA ANALYSIS”. The purpose of this study was to examine the experiences and responses of Canadians to interventions on Twitter using the Kübler-Ross Change Curve (KRCC) during the first six months of the pandemic. Tweets were analyzed using sentiment analysis, thematic content analysis, and KRCC. The findings highlight that many Canadians attempted to adapt to the changes but had a predominantly negative outlook on policies due to the financial and social repercussions. The work seems novel. However, the presentation of the paper needs improvement. It is suggested that the authors make the necessary changes/updates to their paper as per the following comments:
- The relevance of using the Kübler-Ross Change Curve (KRCC) isn’t outlined. Why wasn’t a multilabel classifier developed to detect the six basic emotional states such as happiness, sadness, anger, frustration, disgust, and anxiety? How does this KRCC approach work? Does it use a bag of words model to detect various stages: shock, denial, frustration, depression, experimentation, decision, and integration?
RESPONSE:
We thank the reviewer for asking for this precision. This has now been expanded upon in the revised manuscript.
The following details have been included in the manuscript to better address these questions:
The relevance of the KRCC rather than another multilabel classifier is that while a sentiment analysis was already applied to the tweets, to further understand not just the emotional state of Twitter users but also how Twitter users were adjusting to physical distancing measures, KRCC was used to determine how well Canadians were adapting to change and whether they experienced economic, social or mental impacts from the physical distancing measures based on if they reached the desired stage of integration within the KRCC.
The study uses a bag of words model in NVIVO 11 in order to tokenize the tweet into words and detect the number of times those words were used. Using the common words contained in the tweets, prominent themes were pulled as well as the sentiments associated with those themes. Using the bag of words model, each tweet was assigned to the appropriate stage(s) of the KRCC based on how well it fit each stage. The study combines sentiment analysis, thematic content analysis and KRCC of tweets over six months. This shows how Twitter users were interacting with changes from the pandemic, the major themes and their sentiment towards those themes to understand what are potential causes of stress or negative impacts and what interventions may have helped people adapt better.
- The purpose of this study was to analyze the statements that Canadians made on Twitter regarding their experiences with the perceived threat of COVID-19 and the non-pharmaceutical interventions implemented during the pandemic. Why does this study focus on Canada and not some of the other countries, such as USA, India, etc., which had a significantly high number of COVID-19 cases as compared to Canada?
RESPONSE:
The following details have been included in the manuscript to better address these questions:
Canada was the focus of this study compared to other countries that may have had a higher number of COVID-19 cases given the more restrictive approach to COVID-19 transmission reduction Canada took compared to other countries. We were interested in seeing how these restrictive measures may have unintended social, economic and mental impacts on those effected. These precisions have been added to the manuscript.
- In Section 1.2, the authors briefly review a couple of works that focused on the analysis of Tweets related to COVID-19. Reviewing just 2 works isn’t sufficient as there have been several works done in this field since the outbreak of COVID-19, such as https://doi.org/10.3390/covid2080076 and https://doi.org/10.1111/josi.12535. Consider citing these recent papers as well as at least 2-3 more papers in this review section.
RESPONSE:
We have applied your suggestions and cited more recent papers:
Researchers agree that social media is the medium through which people share their lives. It has become the ultimate “online community”, whereby communication, information, news, views, opinions, perspectives, ideas, knowledge, feedback, and experiences related to pandemics, epidemics, viruses, and disease are discussed by the public at large (Katz & Nandi, 2021; Ng et al, 2022; Thakur & Han, 2022; Wiederhold, 2020).
- Section 4.2 should be rewritten to clearly discuss the steps for data mining. Specifically, please discuss how these tweets were mined. Was the standard search feature or the advanced search feature of Twitter used, and why?
RESPONSE:
The following details have been included in the manuscript to better address these questions:
The Advanced search feature was used because it allowed the researchers to narrow search using the keywords as well as by a specific date range. Once the tweets that fit those criteria were returned, these were extracted and analyzed using Microsoft Excel. From there the inclusion criteria was then applied to the tweets and those that did not fit into the four criteria (tweets from Canada, at least one hashtag from the COVID-19 category and one other category, tweets in English, and tweets dated from March 2020 to August 2020) were excluded.
- The authors state – “Tweets were manually geolocated”. Please discuss the specifics of this process. Did the authors search for the location information of each Twitter user? How did they take into account that some of these Tweets could not be from real users (i.e., the Tweets may have been written by bots)?
RESPONSE:
The following details have been included in the manuscript to better address these questions:
Tweets were manually geolocated by reviewing tweets and Twitter user profiles for location information to ensure that Twitter users were living in Canada and impacted by Canadian policies related to COVID-19. During the data mining process, tweets that did not fit all the inclusion criteria were excluded. In addition, the tweets and Twitter user profiles were further reviewed and excluded if the contents of the tweet or user profile appeared to belong to a Twitter bot.
- The sample size (from Figure 1) seems very less. Consider increasing the number of search words to repeat this study on a greater sample size.
RESPONSE:
A preliminary search had been conducted prior to agreeing on the current search terms. Additional search words/terms actually end up decreasing our population of relevant tweets as we were searching for tweets that crossed at least two categories; we also wanted to ensure that the tweets and users fit within the inclusion criteria listed in the methods section.
- In Section 5, the authors discuss the percentage of negative tweets. Classifying tweets into positive and negative sentiment classes isn’t new, and there have been several research works done in this field that focused on a similar classification of Tweets. Please rewrite this section to clearly highlight why this approach should be considered novel.
RESPONSE:
In order to address the reviewer’s comment, the following was added to section 5:
Researchers agree that positive and negative sentiments interact (Lindgren, 2012). A sentiment analysis on YouTube showed that many negative comments originate from users who are more inclined to post hateful comments without constructive criticism (Lindgren, 2012). Our findings suggest that it is often the case with Tweets. Negative comments are often completely disconnected from the actual content of a video, without offering any constructive criticism to the author, so that the latter may improve his or her technique (Lindgren, 2012). In the context of our analysis, it is evident that even though individuals bear the brunt for most negative Tweets, organizations, the media, and researchers are also among those unleashing such sentiments.
Lindgren, S. (2012). “It took me about half an hour, but I did it!” Media circuits and affinity spaces around how-to videos on YouTube. European Journal of Communication, 27, 152–170
- Is the dataset of Tweets available on a dataset storage/search platform such as IEEE Dataport so that the study may be repeated by other researchers in this field?
RESPONSE:
The dataset is not currently available on a dataset storage/search platform, simply due to a bit of ignorance on the part of the authors on how to actually use these. We will research the suggested IEEE Dataport and make the dataset available.
Reviewer 2 Report
Interesting paper of a twitter analysis in pandemic context, but need some corrections:
* The reference style is no accord to the journal guidelines, please go to the author instructions for checking this.
* May be you could add more references (5 to 7).
* In the methodology section you need to specify some ítems like study design, sampling and population, and variables.
* The results/discussion section in original articles goes separately (IMRAD format), I strongly recommend to change this.
Author Response
Interesting paper of a twitter analysis in pandemic context, but need some corrections:
We would like to thank the reviewer for taking the time to provide thoughtful feedback.
* The reference style is no accord to the journal guidelines, please go to the author instructions for checking this.
RESPONSE
This has been verified and changes made in the manuscript.
* May be you could add more references (5 to 7).
RESPONSE
Six new current references were added to the manuscript to further support the work being presented.
* In the methodology section you need to specify some ítems like study design, sampling and population, and variables.
RESPONSE
The methodology section was expanded to specify the process of generating the population, sample, study design and variables.
* The results/discussion section in original articles goes separately (IMRAD format), I strongly recommend to change this.
RESPONSE
The results and discussion section are now separated to fit the IMRAD format; we feel that the manuscript now reads much better.
Round 2
Reviewer 1 Report
The authors have revised the paper as per all my comments and feedback. I do not have any additional comments at this point. I recommend the publication of the paper in its current form.